# Contextual Memory Bandit for Pro-Active Dialog engagement

## Abstract

An objective of pro-activity in dialog systems is to enhance the usability of conversational agents by enabling them to initiate conversation on their own. While dialog systems have become increasingly popular during the last couple of years, current task oriented dialog systems are still mainly reactive and users tend to initiate conversations. In this paper, we propose to introduce the paradigm of contextual bandits as framework for pro-active dialog systems. Contextual bandits have been the model of choice for the problem of reward maximization with partial feedback since they fit well to the task description. As a second contribution, we introduce and explore the notion of memory into this paradigm. We propose two differentiable memory models that act as parts of the parametric reward estimation function. The first one, Convolutional Selective Memory Networks, uses a selection of past interactions as part of the decision support. The second model, called Contextual Attentive Memory Network, implements a differentiable attention mechanism over the past interactions of the agent. The goal is to generalize the classic model of contextual bandits to settings where temporal information needs to be incorporated and leveraged in a learnable manner. Finally, we illustrate the usability and performance of our model for building a pro-active mobile assistant through an extensive set of experiments.

## 1 Introduction

Dialog agents are about to become the interface of choice for eyeless interactions. The recent progress in the different building blocks of conversational agents have started to make such interfaces usable by the general public in large variety of situations. Furthermore, a large set of APIs has currently been introduced to design task-specific agents through graphical and textual development interfaces. Such interfaces allow even a non-expert to design dialog interfaces which is another reason for the current expansion of such a technology.

The majority of agents developed today assumes a reactive type of interaction (Williams and Young, 2007). Indeed, in a task-oriented setting, the conversation is designed to be initiated by the user. We believe that the seamless nature of conversational interfaces in larger spectrum of contexts makes them particularly suitable for proactive service development. More precisely, the agent will have to infer, regarding a set of observed variables, the pertinence of a given suggestion or conversation engagement on its own and learn from user feedback. A large set of examples can be envisioned. For example, a driving assistant could use a voice interface to pro-actively warn about a potential traffic jam ahead. Similarly, a personal assistant agent could suggest a venue regarding the current location of its user and his interests.

A challenge for building such an agent is the absence of full feedback; the agent needs to infer the quality of its pro-active decisions from the partial feedback given by the user. Contextual bandit (Kakade et al., 2008; Chu et al., 2011) is a particularly appropriate framework to formalize and develop effective solutions with possible theoretical guarantees in such a setting. While in a conventional supervised learning paradigm, the learner has full access to a dataset where true labels are provided, in many domains, including many web-based applications, the agent needs to learn from partial feedback. Online advertising placement and personalized news (Li et al., 2010; Tang et al., 2014) are important examples of such domains. The typical setting can be summarized in the following way. First, the agent observes a current context. Then, using this context and other

source of knowledge (past activities of the user, etc.), the system suggests an action. Finally, the user chooses to react (or not) to the suggestion. Crucially, the system is only able to learn from the user reaction to the suggested action. In the context of a pro-active dialog systems, the setting is comparable. An agent will have to choose the timing and the content of the dialog acts based on (a) the current perceived context (b) its knowledge about the user (c) the reward function to optimize. More formally, the purpose of a contextual bandit is to infer a function that maps the observation and available prior knowledge to the expected reward of a candidate action. Beyond the reward function learning, the partiality of feedback also leads to an exploration-exploitation dilemma.

A limitation of classic framework of contextual bandit is the hypothesis that the current context is sufficient support for decision (Lu et al., 2010; Zhou, 2015). As a second contribution, we propose to incorporate memory to the reward estimation function. Two models are proposed; one with a selective memory and another with a differentiable attention mechanism as proposed in Perez and Liu (2016) computed over a, potentially long, history of contexts. As illustration, we show that such models are particularly meaningful for personal assistance use-cases.

**Roadmap:** Section 2 summarizes the state of the art for incrementality and pro-activity in dialog systems. Section 3 recalls the formalization of contextual bandit and some of its current challenges. Section 4 details our proposed models of contextual bandits with differentiable memorization capabilities. Section 5 describes our exploration policies and their usage in the setting of pro-active dialogs. Finally, Section 6 presents experiments conducted in the domain of personal assistance for mobility and situated suggestions.

## 2   PRO-ACTIVE DIALOG ENGAGEMENT

In this section, we address pro-activity and its links to incrementality in dialog systems. After defining the terms we give an overview of both tasks. In the last part, we address evaluation in such a setting of partial feedback.

### 2.1   DEFINITIONS

The problem of pro-active conversation is often studied from two different angles. On a first hand, the question of *incrementality* studies the usage of spontaneous dialog act emission as part of an active comprehension mechanism (Jonsdottir et al., 2008). Indeed, while maintaining a distribution over latent variables composing the state of the current dialog, the dialog system faces three basic choices a) let the user continue to speak b) repeat a term said by the user for implicit confirmation c) ask the user to repeat for disambiguation or explicit confirmation. These problems are currently solved using rule-based systems as in the work of Baumann et al. (2011) or have been formalized as a delayed reward control tasks that can be solved using reinforcement learning as in the work of Khouzaimi et al. (2016).

On the other hand, *pro-active interaction* is defined as the faculty of a conversational agent to spontaneously address the user. Using a variety of observed and inferred variables from the current perceived context or prior knowledge available, the conversational agent needs to choose either to continue to observe silently or to initiate a conversation regarding a certain subject (like suggesting a task to complete, display a reminder or propose a recommendation). In the context of an eyeless interaction, such capability is crucial yet under-studied. To our knowledge, no work of that kind has been carried out specifically in the context of dialog systems. In the next section, we address the questions of partial feedback and exploitation-exploration which are central to this task.

### 2.2   CHALLENGES

The first challenge of pro-active conversation systems is to develop the capability to rely on partial feedback. Assuming a large panel of available subjects to address and variety of possible recommendations to make, supported by a potentially large panel of decision support variables, the system needs to leverage on a partial feedback that will be provided by the user. Indeed, given a proposition made by the agent, the user will only provide a feedback for the given proposition. In such a setting, a proactive conversational agent needs to solve a so-called exploration/exploitation dilemma. On one hand, the agent needs to leverage on the already gathered feedback to choose propositions

that maximize the current expected reward. On the other hand, the agent needs to explore, i.e. to choose under-investigated propositions to gather meaningful information about the user preferences and profile.

Another difficulty is the necessity of the learnt decision model to incorporate potentially long term information from the history of client interactions, activities and their potential relationship to available prior knowledge. In such situations, using a learnable model with fixed-length memory like a Long Short Term Memory (Hochreiter and Schmidhuber, 1997) is challenging. To cope with this difficulty, attention-based models (Bahdanau et al., 2014) have been suggested in the context of Natural Language Understanding. Then, this proposition has been extended to more general cases of Markovian control. In such settings, memory enhanced controllers has been developed in a full reinforcement learning setting that features (delayed) rewards for chosen actions.

In the next section, we propose to formalize the problem of pro-activity in conversational system with a memory-enhanced contextual bandit model. The purpose is to respond to the dual problem of (1) long term dependencies and (2) partial feedback without the necessity of credit assignment learning.

## 3 CONTEXTUAL BANDIT

Formally, a contextual bandit algorithm $A$ proceeds in discrete trials $t = 1, 2, 3, \ldots, T$. In trial $t$:

1. The algorithm observes the current user $u_t$ and a set $A_t$ of arms or actions together with their feature vectors $\mathbf{x}_{t,a}$ for $a \in A_t$. The vector $\mathbf{x}_{t,a}$ summarizes information of both the user $u_t$ and arm $a$, and will be referred to as the *context*.

2. Based on observed payoffs in previous trials, A chooses an arm $a_t \in A_t$, and receives payoff $r_{t,a_t}$ whose expectation depends on the context $\mathbf{x}_{t,a_t}$.

3. The algorithm improves its arm-selection strategy with the new observation, $(\mathbf{x}_{t,a_t}, a_t, r_{t,a_t})$. It is important to emphasize here that no feedback (i.e., no payoff $r_{t,a}$) is observed for unchosen arms $a \neq a_t$.

In the process above, the total $T$-trial payoff of $A$ is defined as $P_A(T) = \sum_{t=1}^{T} r_{t,a_t}$. Similarly, we define the $T$-trial payoff of an oracle that always chooses an arm with the best expected payoff as $P_{A^*}(T) = \sum_{t=1}^{T} r_{t,a_t^*}$, where $a_t^*$ is the arm with maximum expected payoff at trial $t$. The goal is to design $A$ so that the expected total payoff $\mathbb{E}[P_A(T)]$ is maximized. Equivalently, we may find an algorithm $A$ whose expected regret with respect to the optimal arm-selection strategy $A^*$ is minimized. Here, the expected $T$-trial regret $R_A(T)$ of algorithm $A$ is defined formally by

$$R_A(T) = \mathbb{E}\left[P_{A^*}(T) - P_A(T)\right]. \tag{1}$$

An important simplification of the general contextual bandit problem is the classic $K$-armed bandit in which (a) the arm set $A_t$ remains unchanged and contains $K$ arms for all t, and (b) the user $u_t$ (or equivalently, the context $(\mathbf{x}_{t,1}, \ldots, \mathbf{x}_{t,K})$ ) is the same for all t. Since both the arm set and contexts are constant at every trial, they make no difference to a bandit algorithm, and so we will also refer to this type of bandit as a *context-free* bandit.

Two limitations can be mentioned to the classic framework of contextual bandit. On a first hand, we might need to model potentially long history of interactions as part of the decision support. For example, in the context of a personal assistant a conversation engagement can be triggered regarding a very recent event like the time-matching of a calendar event or the current GPS location of the user. However, longer patterns from series of events occurred through time can motive a suggestion. In such situation, it is difficult to define a priori the necessary time windows of observations to consider. On the other hand, we might explicitly incorporate previous success into an episodic memory and used it as decision support to the reward estimation function. In the next section, we address such problem by proposing learnable parametric reward function that incorporates a differentiable memorization mechanism.

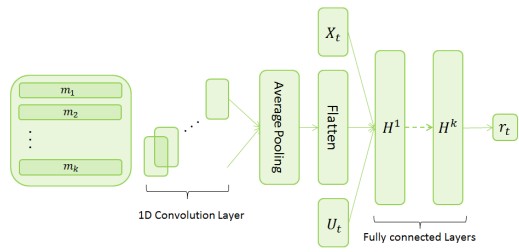

Figure 1: Convolutional Selective Memory Network.

# 4 MEMORY ENHANCEMENT

Memory-enhanced neural networks have recently regained popularity in general reinforcement learning (Hausknecht and Stone, 2015; Sorokin et al., 2015; Oh et al., 2016). More precisely, recurrent neural networks have been used to automatically accumulate observations through time and learn to use a memory to support decision making in a differentiable manner. To our knowledge, such a development is yet to be pursued in the specific domain of contextual bandits. In this section, we propose two reward function models that incorporate such a memorization capability.

## 4.1 CONVOLUTIONAL SELECTIVE MEMORY NETWORK

Our first model is depicted in Figure 1. The Convolutional Selective Memory Network, CSMN, merges three information sources. First, $X_t$ and $U_t$ correspond respectively to the features of the context $X_t$ and the user $U_t$ observed both at time $t$. In addition, a record of the last $K$ successful engagements of the agent with the corresponding user are stored into a list of memory blocks $M_{u_t} = \{m_1, \ldots, m_K\}$. Such memory blocks corresponds to (at most) $K$ latest contexts $X_s$, $(s < t)$ for which the rewards $r_s$ for $U_t$ have been greater than a threshold $\gamma \in \mathbb{R}$.

A 1-dimensional convolutional layer followed by an average pooling layer is used to aggregate meaningful higher level features from these past successful contexts. The model architecture is inspired by Kim (2014). Let $m_i \in \mathbb{R}^k$, k be the k-dimensional vectors describing the individual contexts. The memory is represented as

$$m_{1:n} = m_1 \oplus m_2 \oplus \ldots \oplus m_n, \tag{2}$$

where $\oplus$ is the concatenation operator. Let $m_{i:i+j}$ refer to the concatenation of observations $m_i, m_{i+1}, \ldots, m_{i+j}$. A convolution operation involves a bank of filters. Each filter $\theta \in \mathbb{R}^{hk}$ is applied to a window of $h$ observations to produce a new feature. For example, a feature $c_i$ is generated from a window of stored observations $m_{i:i+h-1}$ by

$$c_i = f(\theta^T m_{i:i+h-1} + b), \tag{3}$$

with $b \in \mathbb{R}$ is a bias term and f is a non-linear function such as the hyperbolic tangent. This filter is applied to each possible window of observations stored into the memory $\{m_{1:h}, m_{2:h+1}, \ldots, m_{n-h+1:n}\}$ to produce a feature map

$$c = [c_1, c_2, \ldots, c_{n-h+1}], \tag{4}$$

with $c \in \mathbb{R}^{n-h+1}$. We then apply an average pooling operation over the feature map and take the average value $\hat{c} = \frac{1}{n-h+1} \sum_{i=1}^{n-h+1} c_i$ as the feature corresponding to this particular filter. The model can be interpreted as a late fusion of old contexts with a current candidate context and user features. The drawback of late fusion is that the memories are not specifically selected regarding the current context. To this purpose, we propose an adaptation of a recently proposed attention-based model presented in the next section.

## 4.2 CONTEXTUAL ATTENTIVE MEMORY NETWORK

Our second model is strongly inspired by the Gated End-to-End Memory Network architecture, introduced by Perez and Liu (2016) as an extension to the work of Sukhbaatar et al. (2015a). It

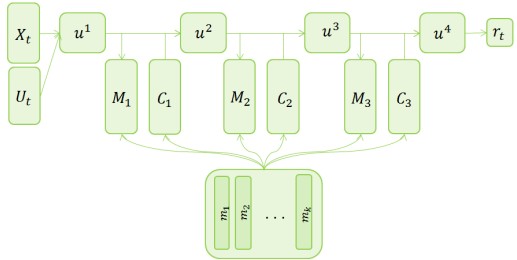

Figure 2: Contextual Attentive Memory Network with 3 hops.

consists of two main components: supporting memories and expected reward prediction. Supporting memories are in turn comprised of a set of input and output memory representations with memory cells. The input and output memory cells, denoted by $m_i$ and $c_i$, are obtained by transforming the input context $x_1, \ldots, x_n$ (i.e a set of observations) using two embedding matrices $A$ and $C$ (both of size $d \times |V|$ where $d$ is the embedding size and $|V|$ the observation size) such that $m_i = A\Phi(x_i)$ and $c_i = C\Phi(x_i)$ where $\Phi(\cdot)$ is a function that maps the input into a vector of dimension $|V|$.

Similarly, the current context $x$ concatenated with the information regarding the user are encoded using another embedding matrix $B \in \mathbb{R}^{d \times |V|}$, resulting in a context embedding $u = B\Phi(q)$. The input memories $\{m_i\}$, together with the embedding of the context $u$, are utilized to determine the relevance of each of the history embedded into the memory blocks in the context, yielding in a vector of attention weights

$$p_i = \text{softmax}(u^\top m_i), \tag{5}$$

where $\text{softmax}(a_i) = \dfrac{e^{a_i}}{\sum_i e^{a_i}}$. Subsequently, the response $o$ from the output memory is constructed by the weighted sum:

$$o = \sum_i p_i c_i. \tag{6}$$

For more difficult tasks requiring multiple supporting memories, the model can be extended to include more than one set of input/output memories by stacking $K$ memory layers, often called hops, so that the $(k+1)^{\text{th}}$ hop takes as input the output of the $k^{\text{th}}$ hop:

$$T^k(u^k) = \sigma(W_T^k u^k + b_T^k) \tag{7}$$
$$u^{k+1} = o^k \odot T^k(u^k) + u^k \odot (1 - T^k(u^k)), \tag{8}$$

where $\odot$ denotes the Hadamard-product, $\sigma(x) = (1 + e^{-x})^{-1}$, $W_T^k$ and $b^k$ are the hop-specific parameter matrix and the bias term for the $k^{\text{th}}$ hop, and $T^k(x)$ is the transform gate for the $k^{\text{th}}$ hop.

Two types of weight tying schemes Sukhbaatar et al. (2015b) have been investigated for $W_T^k$ and $b_T^k$:

1. **Global:** all the weight matrices $W_T^k$ and bias terms $b_T^k$ are shared across different hops, i.e., $\forall k,l \in \{1,\ldots,K\}\ W_T^k = W_T^l$ and $b_T^k = b_T^l$.
2. **Hop-specific:** each hop $k$ has its specific weight matrix $W_T^k$ and bias term $b_T^k$, and they are optimized independently.

As the final step, the prediction of the expected reward $r$ for the input context, is computed by

$$r_t = \left( u_K^\top W' \right),$$

where $W' \in \mathbb{R}^d$ is part of the parameters for the model to learn. Such enhancement allows to deal with the non-Markovian hypothesis in the context definition. In Section 6 the proposed model is used to implement a goal-oriented proactive dialog agent.

---

**Algorithm 1** Thompson Sampling

Define $D = \{\}$
**for** $t : 1, \ldots, T_{max}$ **do**
    receive context $x_t$
    Draw $\theta$ from posterior distribution $P(\theta|D)$
    Select arm $a_t = \text{argmax}_a E(r|x_t, a, \theta_t)$
    Receive reward $r_t$
    $D = D \cup (\{x_t, a_t, r_t\})$
**end for**

---

## 5 EXPLORATION POLICIES

A key challenge in bandit problems is the optimal balancing of exploration and exploitation. To minimize the regret in Eq. 1, an algorithm $A$ exploits its past experience to select the arm that appears the best. On the other hand, this seemingly optimal arm may in fact be suboptimal, due to incompleteness of $A$'s knowledge. In order to make informed decisions, $A$ has to also explore by sometimes choosing seemingly suboptimal arms in order to gather more information about them (c.f., step 3 in the bandit process defined in the previous subsection). This exploration can increase short-term regret since some suboptimal arms may be chosen, but on the other hand, obtaining information about the arms' average payoffs will later enhance $A$'s capability to reduce long term regret. Clearly, neither a purely exploring nor a purely exploiting algorithm works best in general but a good trade-off is needed.

In this work, we use Thompson sampling which is a popular Bayesian heuristic to achieve a successful exploration/exploitation trade-off Chapelle and Li (2011); Agrawal and Goyal (2013). Let $D$ be the set of past observations $(x_t, a_t, r_t)$, where $x_t$ is the context in which the arm $a_t$ was pulled yielding the reward $r_t$. Thompson sampling assumes a likelihood function $P(r|a, x, \Theta)$ paramerized by $\Theta$ for the reward. If we denote the true parameters by $\theta^*$, we would ideally choose an arm that maximizes the expected reward $\text{argmax}_a E[r|a, x, \theta^*]$, but since we dont know the true parameters we describe our belief about it by a prior distribution $P(\Theta)$, and then based on the observed data $D$, update this belief by the Bayes rule $P(\Theta|D) \propto P(\Theta) \prod_{t=1}^{T} P(r_t|x_t, a_t, \Theta)$. Now if we just wanted to maximize the immediate reward, we would choose an arm that maximizes $E[r|a, x] = \int E(r|a, x, \Theta) P(\Theta|D) d\Theta$, but in an exploration/exploitation setting, Thompson sampling advises to select an arm $a$ according to its probability of being optimal, i.e., with probability

$$\int \mathbb{I}[\mathbb{E}(r|a, \Theta) = \underset{a'}{\text{argmax}} E[r|a', \Theta]] P(\Theta|D) d\Theta. \tag{9}$$

In practice, we do not need to calculate the integral but it suffices to draw a random parameter $\theta$ from posterior distribution $P(\Theta|D)$ and then select the arm with the highest expected reward $E[r|a, x, \theta]$. The general framework of Thompson sampling is described in Algorithm 1.

### 5.1 GOAL-ORIENTED PROACTIVE DIALOG POLICY

In a pro-active dialog system, an agent has to periodically choose whether and what to interact to the user at time $t$. Its choices are picked among elibigible actions over the current context $x_{t,a_t} \in X$ and a history of context variables $x_1, \ldots, x_{t-1} \in H$. To do so, the agent needs to learn a function $f_\theta : (H, X_A) \to \mathbb{R}$ which estimates the expected reward associated to a candidate contextualized action and memorized history.

Following the paradigm of contextual bandits, the agent will estimate the expected reward of an action through the feedback of the user. The goal is to accumulate positive rewards for the chosen conversational engagements. Feedback can be explicit like a positive or a negative response, implicit or even no response at all. An example of an implicit positive feedback might be the observed change in travel trajectory in the case of the suggestion of place during a travel. In the next section, we give a detailed description of the experimental setting and evaluation of our models.

Table 1: Top representative bi-grams for a set of 15-topics extracted from reviews. Each topic corresponds to a latent centre of interest of the user and observed description of the places.

| |
|---|
| dim_sum , late_night , 20_minutes , took_long , large_groups , wednesday_night , sweet_potato_fries , potato_fries , sweet_potato , beer_pong |
| happy_hour , russ_daughters , tables_available , drinks_happy_hour , drinks_happy , group_people , lyft_credit , open_table , portions_huge , lamb_shank |
| live_music , beers_tap , gluten_free , little_italy , small_cozy , cheap_drinks , main_course , drinks_bar , cooked_perfectly , group_friends |
| saturday_night , dining_experience , prix_fixe , selection_drinks , stay_away , small_portions , beca_friend , huge_selection , portions_small , sat_bar |
| french_toast , date_night , hip_hop , don_come , outdoor_seating , cozy_little , second_floor , date_spot , ve_tried , marinara_sauce |
| east_village , dining_area , beer_bar , old_school , financial_district , statue_liberty , long_wait , craft_beer , goat_cheese , hole_wall |
| selection_beers , small_plates , cool_spot , grab_meal , drinks_music , healthy_lunch , fresh_air , soup_dumplings , lobster_roll , breath_fresh |
| fried_chicken , hang_friends , tuna_melt , peanut_butter , grab_drink , cocktail_bar , free_wifi , week_lunch , eggs_benedict , cheese_plate |
| sitting_bar , cup_coffee , bloody_mary , took_care , little_slow , fish_dishes , makes_feel , cold_brew , course_tasting , wide_variety |
| avocado_toast , low_key , chocolate_cake , ve_seen , happy_hour , fried_egg , large_group , new_people , meet_new , ho_ketchup |
| coffee_shop , free_lyft , bar_bartender , pasta_dish , fish_chips , cool_atmosphere , cute_little , atmosphere_music , bartender_drinks , brooklyn_bridge |
| happy_hour , wine_list , went_lunch , friday_night , hour_specials , happy_hour_specials , people_watching , bars_city , bread_pudding , people_don |
| 10_minutes , bar_tenders , dining_room , bar_area , fun_atmosphere , minutes_later , atmosphere_drinks , soup_salad , birthday_party , steak_sandwich |
| ice_cream , just_right , make_feel , feel_welcome , little_cafe , wonderful_experience , did_job , mac_cheese , 15_20 , bar_bit |
| beer_selection , dive_bar , wine_bar , wine_selection , free_lyft , neighborhood_bar , irish_pub , irish_irish , little_expensive , walked_away |

## 6 EXPERIMENTS

In this section, we describe the simulation platform and the experiments we developed to study the performance of the proposed models.

### 6.1 MOBILITY AGENT PLATFORM

The overall context of our simulations is situated recommandations. The purpose of the platform is to simulate the spontaneous engagement of a personal assistant with its user regarding situated recommandations near its geographic location. The platform simulates a city where a user follows a series of trajectories composed of waypoints. Situated contextual observations are used to simulate places of interest that will be suggested to user when they arrive at their vicinity. In summary, the goal of the agent is to propose actions regarding the inferred latent variables of the user representing its interests.

In order to enrich the simulation platform with realistic elements of information, we extracted from Google Map Place API 6000 reviews from the city of New York corresponding to four types of places: *restaurant, bar, pub, cafe*, for a total of 5000 places. A topic model based on Latent Dirichlet Allocation (Blei et al., 2003) was computed from these reviews. The topic distributions were used (1) as latent variable representing the user interests and (2) as latent variable representing the places. In order to leverage on reviews to extract meaningfull topics to characterize different types of places, we suppress to the reviews a lexicon of emotion related words and consider only 2-grams as vectorization of the reviews. Table 1 gives a sample of the latent topics extracted.

The topic model is used to sample keywords that act as observations of places. Table 2 gives examples of places with their corresponding latent topics and observed keywords. A simulated user is defined by a distribution over the latent topics which correspond to its centre of interest. For the places, a distribution of latent topics is drawn from the topic model and a series of observable keywords are sampled from the LDA topic model accordingly. In such a way, the personal assistant has access to a set of topic related keywords of each place of the simulated city. Its goal is to infer the latent centres of interest of the user by recommanding corresponding places as the user passes nearby. The reward $r$ associated to a user latent centre of interest vector $\mathbf{u}$ and the latent topic of a given place $\mathbf{x}$ is computed by: $r = u^T x + \varepsilon$ with $\varepsilon \sim \mathcal{N}(0, \sigma^2)$ and $\sigma \in \mathbb{R}$ is the predefined variance parameter of a Gaussian distributed noise.

Algorithm 2 describes the course of a simulation. The users performs a series of travels composed with waypoints into the city. At each waypoint of the trajectory, the agent has to decide whether to make a recommandation regarding a place nearby the current location of the user. If a recommandation is decided, the corresponding $r$ is computed and presented as instantaneous reward to the personal assistant. Finally, the agent learns from its experience the expected reward of such context. In practice, such reward could be formulated in a real environment as the explicit acceptance by the user or refusal. The addition of the noise to the reward function helps to simulate the challenge of interpretation of a user feedback in a realistic setting. In summary, the dialog act of the agent consists of proposing a place near the current user position.

---

**Algorithm 2** Simulation Dynamics

---

$H = \{\}$
**for** $e : 1, \ldots, TotalNumberOfEpochs$ **do**
    sample user $u_e \in U$
    sample trajectory $p_1, \ldots, p_T$
    **for** $t : 1, \ldots, T$ **do**
        extract the $n$ places $(a_1, \ldots, a_n)$ closest to $p_t$
        Draw $\theta$ from Thompson sampling
        Choose $a^* = \text{argmax}_{a_i} f_\theta(u_e, a_i)$
        Observe reward $r_t$
        $H = H \bigcup \{(u_t, a^*, r_t)\}$
    **end for**
**end for**

---

Table 2: Four places with main latent topics and observed sampled keywords.

| Topics | Keywords |
|---|---|
| 8 | bloody_mary  canned_tuna  cheap_beer  chips_typical_nyc come_friends  cool_spot  east_village  french_toast  gigantic_portion  gluten_free  happy_hour  house_wine kind_hospitality little_expensive olive_oil ordered_burger outdoor_seating sandwich_steak_salad second_floor spot_brunch standard_bar strawberry_shake usual_bar vegetarian_options veggie_burger |
| 16 | attentive_drinks  awesome_spot  bar_bit  cafe_mocha coffee_shop  cool_bar  fish_dishes  french_toast  library_hotel_vacation long_wait pretty_expensive relaxed_bar scrambled_eggs  seating_inside  selection_wine  spot_city sunday_night water_refills went_sunday |
| 11 | appetizer_entree  bars_city  beer_selection  beet_relish cozy_spot  dance_floor  downtown_tavern  fresh_squeezed fried_egg  jazz_band  large_group  red_wine  small_beer small_space  steak_tartare  took_forever  usually_crowded ve_nyc vegan_options went_week wine_bars |
| 6 & 8 | atmosphere_pleasant  authentic_japanese  bar_big  bathrooms_clean cheese_steak craft_beer dim_sum drinks_decent free_wifi  hang_friends  happy_hour  little_gem  lot_fun pint_guinness  poached_egg  salad_pattern_did  seat_bar seating_available soup_dumplings sunday_night |

For the experiments, 500 places are sampled over a squared area composed with 300 waypoints. User's trajectories are computed on-the-fly over sampled departure and arrival locations using a shortest path algorithm. The nearby candidate locations at each waypoint are determined using a fixed radius around the user location. A total of 50 users have been sampled for each experiment. Finally, each experiment consists in a total of 1000 trajectories which correspond to the number of epochs. Each experiment is relaunched 5 times for variance estimation. Regarding the learning algorithms, hyper-parameters have been determined by cross-validation. First, the CSMN model uses 64 filters of width 3 in its convolution layer and 4 hidden layers in its fully connected part with 100 hidden units each. Then, the CAMN model uses 3 hops with an embedding size of 30. The hop-specific weight tying schema has been adopted. The last part of the section presents a series of experimental results of the proposed models against a state of the art contextual neural network and an random policy.

## 6.2 RESULTS

Figure 3 plots the cumulative sum of rewards for the four considered methods. Regarding the baselines, Fully Connected Neural Network, FFCN, is a multi-layer perceptron which merges the context variable $x_t$ and the user features $u$ into the input layer of the model and computes the expected reward of this corresponding context. Two main observations can be provided.

**Selective memory mechanism improves performance over neural contextual bandit**. Indeed, having a fixed sized memory over the past successfully selected places is a useful addition as decision support. The convolutional memorization mechanism seems effective to such purpose.

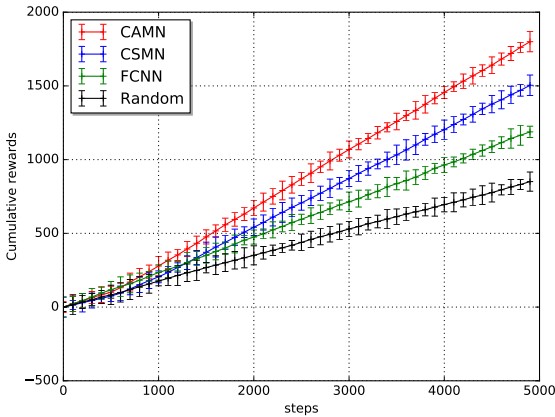

Figure 3: Mean cumulative sum of rewards.

**Attention-based models of the contextual memory bandit is beneficial**. The capability to have an unbounded memory with a differentiable attention mechanism improves the observed cumulative rewards.

## 7 CONCLUSION AND FUTURE WORK

In this paper we present a novel model for designing pro-active dialog agents. The model is in charge of deciding the timing and the nature of an interaction. We propose to formalize this problem using the framework of contextual memory bandit which is an extension of the classic contextual bandit formalism with the capability of handling variable length history through the usage of a differentiable attention model. We experiment our models in the context of a simulated platform of personal agent which is in charge of emitting spacio-temporally motivated recommendations. We believe this framework constitutes a strong baseline for future research in the domain of pro-active dialog agents as conversational interfaces become popular. Finally, we plan two research topics for future work. On the first hand, we think the hypothesis of contextual bandit can be questioned. Indeed by allowing exploration, the agent might end up changing the preferences of its user. In fact, this stationary hypothesis over the latent variables describing the user, but also the places, are limitation that we plan to address. On the other hand, we want to integrate external knowledge base as part of the memory of our decision models.

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
