# OpenReview forum: "Contextual memory bandit for pro-active dialog engagement"
_ICLR.cc/2018/Conference — Reject_

### Official Review · AnonReviewer3 · 2017-11-21
**Contextual Memory bandit for pro-active mobiliy agent dialog interfaces**

**Rating:** 2
**Confidence:** 5

**Review:**

This article propose to combine a form of contextual Thompson sampling policy with memory networks to handle dialog engagement in mobility interfaces.

The idea of using contextual bandits (especially Thompson sampling) instead of state of the art approximate RL algorithms (like DQN, AC3, or fittedQ) is surprising given the intrinsic Markovian nature of dialog. Another key difficulty of dialogs is delayed feedback. Contextual bandits are not designed to handle this delay. I saw no attempt to elaborate on that ground.
One possible justification, however, could be the recent theoretical works on Contextual Decision Processes (See for instance the OLIVE algorithm in "Contextual Decision Processes with Low Bellman rank are PAC learnable" by Jiang et al. @ NIPS 2016). A mix of OLIVE, memory networks and Thompson sampling could be worth studying.

The article is however poorly written and it reflects a severe lack of  scientific methodology : the problem statement is too vague and the experimental protocol is dubious.

The dialog engagement problem considered is a special case of reinforcement learning problem where the agent is given the option of "doing nothing". The authors did not explain clearly the specificity of this dialog-engagement setting with regard to other RL settings. For instance why not evaluate their algorithm on video-games where doing nothing is an option?

The authors introduce the notion of regret on page 3 but it is never used after.
It is unclear whether the problem considered here is purely online (cold start) or off-line (repeated training on a simulator).

In order to obtain a decent experimental protocol for their future submissions I suggest that the authors provide:
- both offline (repeated training on simulator) and online (cold-start) experiments;
- at least more than one dialog simulation scenario (in order to convince you reader that your experimental result is more than a side-effect of your simulator bias);
- at least a few baselines with state of the art deep and shallow RL algorithms to justify the use of a new method.

---

### Official Review · AnonReviewer2 · 2017-11-27
**Not enough detailled implementation. Not convincing experiments**

**Rating:** 3
**Confidence:** 4

**Review:**

The paper "CONTEXTUAL MEMORY BANDIT FOR PRO-ACTIVE DIALOG ENGAGEMENT" proposes to address the problem of pro-active dialog engagement by the mean of a bandit framework that selects dialog situations w.r.t. to the context of the system. Authors define a neural archiecture managing memory with the mean of a contextual attention mechanism.

My main concern about this paper is that the proposal is not enough well described. A very large amount of technical details are missing for allowing the reader to understand the model (and reproduce the experiments). The most important ones are about the exploration policies which are not described at all, while it is a very central point of the paper. The only discussion given w.r.t. the exploration policy is a very general overview about Thompson Sampling. But nothing is said about how it is implemented in the case of the proposed model. How is estimated p(\Theta|D) ? Ok  authors give p(\Theta|D) as a product between prior and likelihood. But it is not sufficient to get p(\Theta|D), the evidence should also been considered (for instance by using variational inference). Also, what is the prior of the parameters ? How is distributed r given a,x and \Theta ?

Also, not enough justification is given about the general idea of the model. Authors should give more intuitions about the mechanism they propose. Figure 2 should be able to help, but no reference to this figure is given in the text, so it is very difficult to extract any information from it. Authors only (roughly) describe  the architecture without justifying their choices.

At last, the experiments really fail at demonstrating the relevance of the approach, as only questionable artificial data is used. On the first hand it appears mandatory to me to consider some (even minimal) experiments on real data for such proposal. On the other one, the simulated data used there cannot correspond to cues to validate the approach since they appear very far from real scenarios: the trajectories do not depend on what is recommended. Ok only the recommended places reveal some reward but it appears not as a sufficiently realistic scenario to me. Also, very too few baselines are considered: only different versions of the proposal and a random baseline are considered. A classical contextual bandit instance (such as LinUCB) would have been a minimum.

Other remarks:
      - the definition of q is not given
      - user is part of the context x  in the bandit section but not after where it is denoted as u.
      - the notion of time window should be more formally described
      - How is built the context is not clear in the experiments section

---

### Official Review · AnonReviewer1 · 2017-11-28
**This paper attempts to use contextual bandits for a dialog system. The paper is not clear about how exactly the problem is being mapped to the contextual bandit framework. Similarly, the Thompson sampling algorithm is used, but there is no mention of a posterior or how to sample. Furthermore, the lack of systematic experiments and ablation studies casts doubts on the entire framework.**

**Rating:** 3
**Confidence:** 4

**Review:**

This paper attempts to use contextual bandits for a dialog system. The paper is not clear about how exactly the problem is being mapped to the contextual bandit framework. Similarly, the Thompson sampling algorithm is used, but there is no mention of a posterior or how to sample. Furthermore, the lack of systematic experiments and ablation studies casts doubts on the entire framework. Below is a detailed review and questions for the authors:
1. Please motivate clearly the need for having a bandit framework. One would imagine that dialog systems have a huge amount of data that can be leveraged for a pro-active service.
2. In the sentence "the agent needs to leverage on the already gathered feedback to choose propositions that maximize the current expected reward". The expected reward or the agent is undefined at this point in the paper.
3. After introducing the contextual bandit problem, please give the explicit mapping of your system to this framework. What do the arms correspond to, how are they related, how is the expected reward computed at each round? Another thing which is not clear is what is the environment? It seems that a recommendation by the dialog system will cause the environment to change, in which case it's not a bandit problem? Isn't it more natural to model this problem as a reinforcement learing problem?
4. In the sentence, "a record of the last K successful engagements of the agent". It is not clear what constitutes a successful engagement. Also, please justify why you are not keeping negative examples in order to learn.
5. Section 5 describes the Thompson sampling algorithm. Again, there is no mapping between the problem at hand and the TS framework. For instance, what is the posterior in this case. How are you sampling it? Are we being Bayesian, if so what is the prior?
6. In the sentence "longer patterns from series of events occurred through time can motive a suggestion", it seems that the problem you are trying to solve involves delayed feedback. Can you use the strategies in [1] over here?
7. For equation 8, please give some intuition. Why is it necessary?
8. In the sentence, "Regarding the learning algorithms, hyper-parameters have been determined by cross-validation.". Isn't the data is being collected on the fly in a bandit framework? What is the cross-validation being done on?
9.One experiment on one dataset does not imply that the method is effective. Similarly, the lack of ablation studies for the various components of the system is worrying.
[1] Guha, Sudipto, Kamesh Munagala, and Martin Pal. "Multiarmed bandit problems with delayed feedback." arXiv preprint arXiv:1011.1161 (2010).

---

### Decision · Program_Chairs · 2018-01-29
**ICLR 2018 Conference Acceptance Decision**

**Decision:**

Reject

**Comment:**

This paper is lacking in terms of clarity and experimentation, and would require a lot of additional work to bring it to the standards of any high quality venue.